# Characterization of Hot Workability in AISI 4340 Based on a 3D Processing Map

**Joonhee Park** [1,†] , **Yosep Kim** [1,†] , **Sangyun Shin** [2] **and Naksoo Kim** [1,*]

1   Department of Mechanical Engineering, Sogang University, Seoul 04107, Korea
2   SBB TECH Inc., Gimpo 10020, Korea
*   Correspondence: nskim@sogang.ac.kr; Tel.: +82-2703-8635
†   These authors contributed equally to this work.

**Abstract:** This study performed high-temperature compression tests at the temperature 900 to 1200 °C and strain rate 0.01 to 10 s$^{-1}$ to characterize the high-temperature deformation behavior of AISI 4340. The constitutive equation of AISI 4340 was expressed using the Arrhenius model and the Zener–Hollomon (Z) parameter. Dynamic Recrystallization (DRX) behavior was evaluated by observing the compressed specimen with Electron Backscatter Diffraction (EBSD). The processing map is based on the dissipation efficiency of the dynamic material model (DMM) and the plastic instability criterion of Ziegler. At strain 0.4, the power dissipation efficiency value is 0.5 or more, and the instable zones are immediately identified through the processing map. The strain, strain rate and temperature data obtained from the FEM simulation of the hot forging process are displayed on the proposed 3D processing map to avoid the flow instability zones and ensure high power dissipation efficiency zones, allowing the operator to control the process's temperature and speed.

**Keywords:** AISI 4340; 3D processing map; forging process control; particle tracking technique; hot deformation behavior; dynamic recrystallization

## 1. Introduction

In manufacturing parts of high-strength, high-temperature materials, process parameters are generally selected based on test methods such as tensile, compression, and bending tests [1,2]. This study used a hot compression test similar to the hot forging process behavior. The compression test affects deformation behavior due to deformation heat and the friction force between the die and the material. In addition, the heat of deformation affects the properties of the product by making the production process complex and non-uniform [3]. Therefore, the flow stress obtained in the compression test must be corrected for deformation temperature and friction.

Causes of defects in plastic deformation include friction between the die and the temperature non-uniformity between the surface and center of the material [4]. In particular, flow behavior of steels depends on the strain, temperature, strain rate, and deformation mode of hot deformation [5–7]. These processes encounter optimization challenges associated with the deformation behavior. In addition, a production process that relies on the operator's experience requires a lot of trial and error and cost burden [8]. Therefore, designing an efficient metal-forming process is necessary to lower the production cost.

The Arrhenius [9] and Johnson–Cook [10] phenomenological models have been widely used for flow stress behavior at high temperatures. The Arrhenius model can be expressed simply with a Zener–Hollomon (Z) parameter. In addition, since compensation for temperature, strain rate, and strain is possible, the flow stress behavior at a high temperature can be accurately described [11,12]. In the case of hot forging, the microstructure is evaluated by softening mechanisms such as dynamic recovery (DRV) and dynamic recrystallization (DRX) [13–15]. The ability of the grain boundary is related to the pinning effect and its

mobility [16]. Recently, the machinability properties of metal materials in hot deformation based on the dynamic material model (DMM) have been used in processing maps that can evaluate the deformation mechanism of the determined region and explain the unstable zones [17,18]. Frost and Ashby developed the first deformation process theory in 1982 to determine whether defects occurred during the process due to process parameters such as strain, strain rate, and temperature [19]. Since then, Prasad has developed a modified process map configured following the principle of DMM [20].

In this study, a method of controlling the temperature and speed of the hot forging process is present by visualizing the deformation history of the nodes inside the material on the 3D processing map space. The compression test results corrected the strain-stress curve for the flow stress by considering the temperature change and friction. A constitutive equation was established through the corrected flow stress. The microstructures were characterized through a SU5000 HITACHI scanning electron microscope with a VELOCITY SUPER EDAX electron backscatter diffraction (EBSD) detector. In addition, a 3D flow instability map was produced to determine the strain rate and temperature during the process to avoid forging defects during high-temperature deformation. A processing map was first proposed by Prasad and Seshacharyulu in 1998 [21]. Existing processing maps plot the power dissipation efficiency and flow instability on a specific strain's temperature and strain rate planes. However, during plastic deformation, strain, strain rate, and temperature change continuously according to the deformation of the material, so the deformation history of the particles was plotted in a 3D processing map space through finite element method (FEM) simulation. Finally, through the 3D processing map, the process temperature and speed were determined according to the strain of the product during the hot forging process.

## 2. Materials and Methods

Table 1 lists the composition of AISI 4340 steel used in this study. Figure 1a shows the specimen of the hot compression test of AISI 4340 material with an original diameter of 10 mm and a height of 15 mm using the Gleeble 3500 simulator (Dynamic System Inc., Austin, Tex, USA). Figure 1b shows the specimens were compressed to true strain 1.0 at a temperature of 900, 1000, 1100, and 1200 °C, strain rate of 0.01, 0.1, 1, and 10 s$^{-1}$. The specimen was heated at the rate of 10 °C/s to the target temperature. Then, the specimen was compressed by maintaining it at a specific temperature for 180 s so that the temperature of the specimen distributes uniformly; finally, it was air-cooled. Figure 2 shows the grain boundary (GB) map of the initial specimen, and the grain size is 24.58 μm.

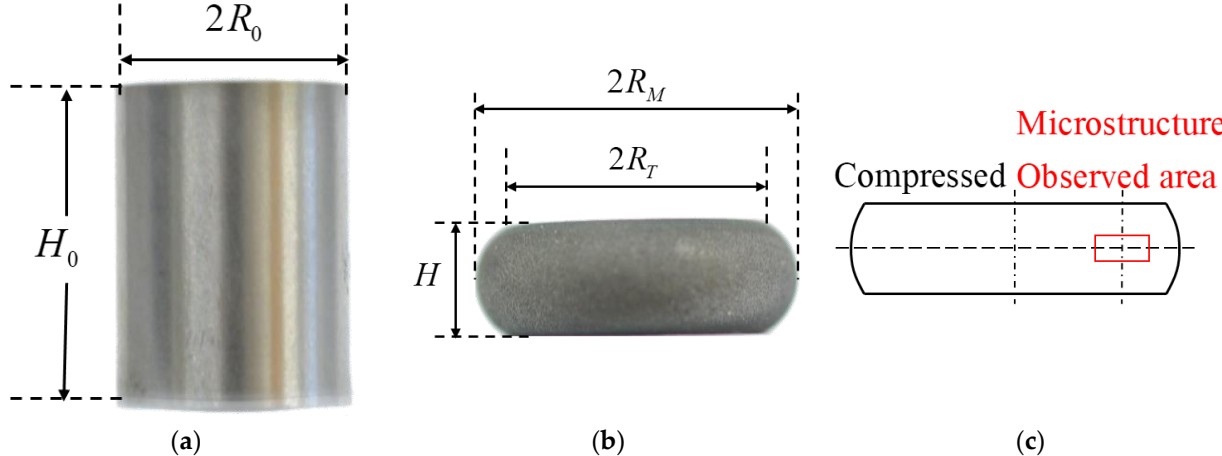

**Figure 1.** Compression test specimen: (**a**) before deformation, (**b**) after deformation, (**c**) the area for microstructure observation.

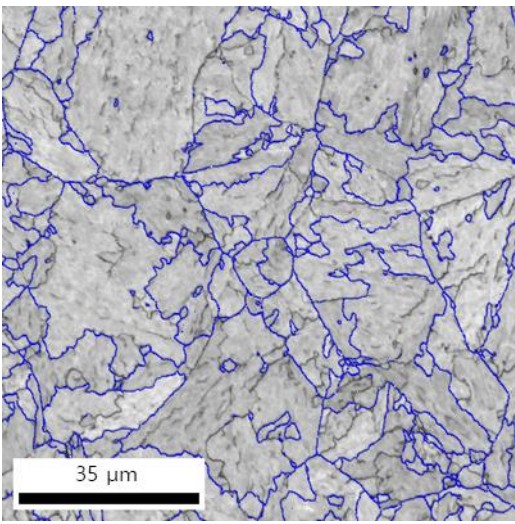

**Figure 2.** GB map of the initial specimen.

**Table 1.** Chemical compositions of the studied steel (mass fraction, %).

| Elements | C | Si | Mn | P | S | Ni | Cr | Mo | Cu | Fe |
|----------|------|------|------|-------|-------|-----|------|------|-----|------|
| Mass fraction (wt. %) | 0.42 | 0.18 | 0.84 | 0.018 | 0.006 | 1.6 | 0.79 | 0.15 | 0.1 | Bal. |

## 3. Results and Discussion

### 3.1. Friction and Temperature Correction

In the hot compression test, tantalum plates were used to reduce the effect of friction. However, the deformation-stress curve obtained through the compression test is different from the actual curve due to the intervention of frictional force. Therefore, the flow stress must be corrected, considering the effect of friction. The effect of friction becomes more evident as strain increases. The effect of friction on the flow stress was corrected using Equation (1) as follows [22,23].

$$\sigma_f = \frac{\sigma}{1 + \left(2/3\sqrt{3}\right)\mu(R_0/h_0)\exp(3\varepsilon/2)} \tag{1}$$

where $\sigma_f$ is the corrected flow stress, $\sigma$ is the measured flow stress, $\varepsilon$ is the measured strain, $r$ is the transient average radius of the specimen during compression, and $\mu$ is the coefficient of friction. The friction coefficient is calculated using the energy method proposed by Ebrahimi et al. [24], with the calculation formula as follows:

$$\mu = \frac{(R_A/H)b}{4/\sqrt{3} - 2b/3\sqrt{3}} \tag{2}$$

where $R_A$ is the average radius after deformation, $H$ is the height of the specimen after deformation, and $b$ is the structure coefficient known by the following formulae:

$$b = 4\frac{\Delta R}{R_A}\frac{H}{\Delta H} \tag{3}$$

$$R_A = R_0\sqrt{\frac{H_0}{H}} \tag{4}$$

The deformation history makes it difficult to directly measure the specimen's top radius ($R_T$) and maximum radius ($R_M$). The final $R_M$ is measured after the compression

test and the approximation of the barreled profile specimens with the arc of a circle ($R_T$) can be determined by the functions as follows:

$$R_T = \sqrt{3\frac{H_0}{H}R_0{}^2 - 2R_M{}^2}$$ (5)

$$\Delta R = R_M - R_T$$ (6)

A thermocouple was attached to the center of the specimen to record temperature data in real time during the compression test. Figure 3 shows the temperature change according to the deformation. In the high-temperature compression test, the temperature change due to the deformation heat differs from the stress at the desired temperature. Therefore, the flow stress corrected by the effect of friction must correct to the desired temperature. The temperature and stress relationship were corrected by linear interpolation to match the set temperature of the isothermal compression test. Figure 4 shows the flow stress corrected by friction and temperature change. The difference between the experimental and corrected flow stress became larger at high strains since the effects of friction and temperature changes became higher as the strain increases.

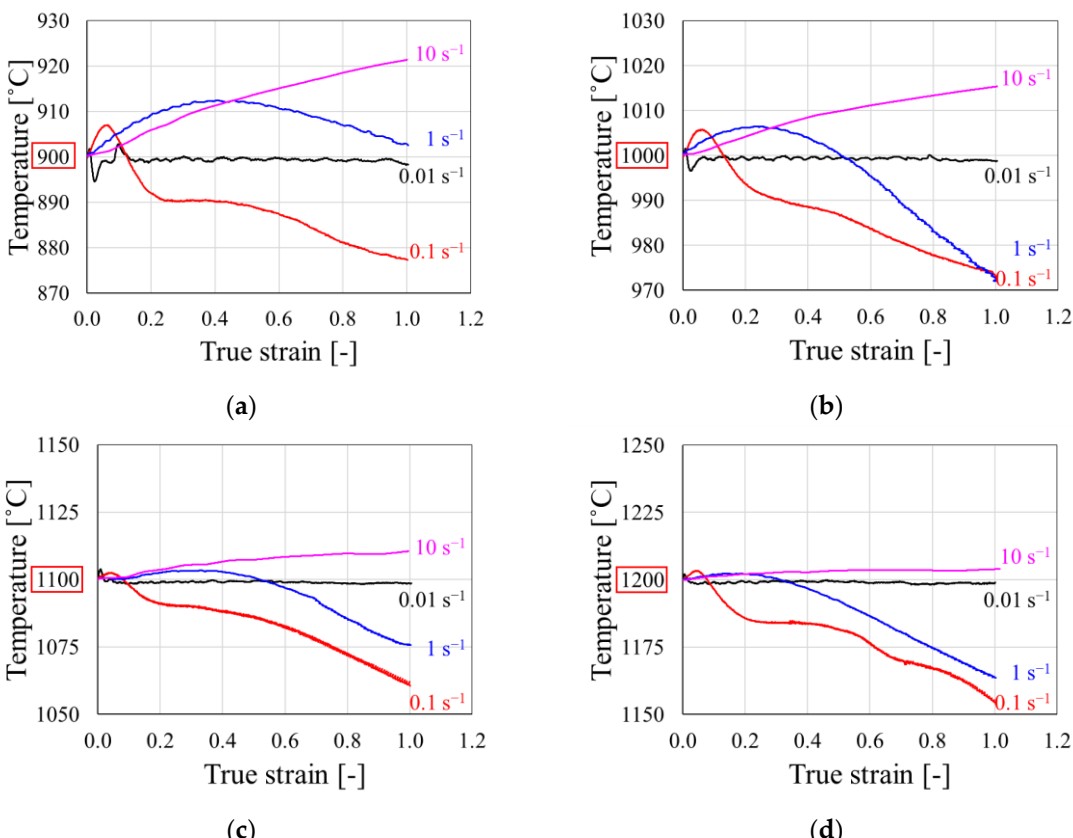

**Figure 3.** Temperature change of specimen during compression test: (**a**) 900 °C, (**b**) 1000 °C, (**c**) 1100 °C, (**d**) 1200 °C.

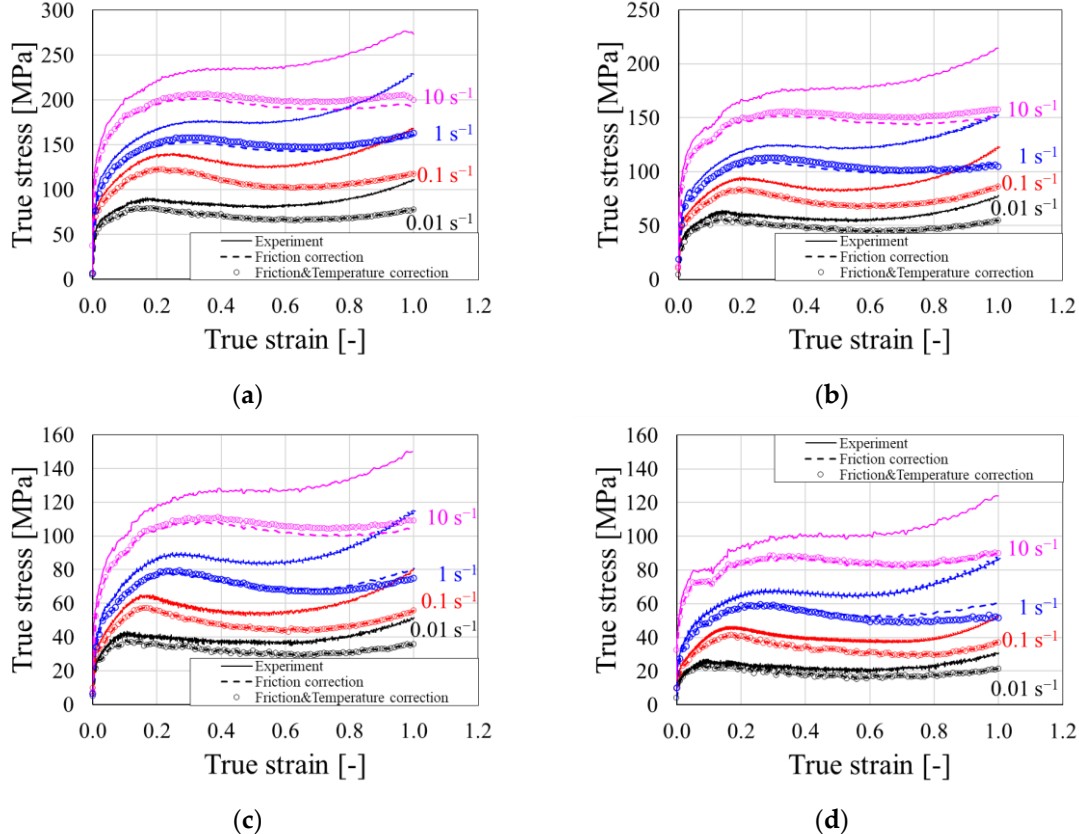

**Figure 4.** True stress-strain curves of the experiment were corrected under different deformation conditions: (**a**) 900 °C, (**b**) 1000 °C, (**c**) 1100 °C, (**d**) 1200 °C.

### 3.2. Derivation of Arrhenius Constitutive Equation

In general, in the hot deformation of the metal, the flow stress is mainly affected by strain rate and temperature. As follows in Equation (7), the Arrhenius constitutive equation proposed by Sellers and McTegart describes the stress range by temperature and strain rate in hot deformation [10].

$$\dot{\varepsilon} = AF(\sigma)\exp\left(\frac{-Q}{RT}\right), \text{ where } F(\sigma) = \begin{cases} \sigma^{n_1}, & \alpha\sigma < 0.8 \\ \exp(\beta\sigma), & \alpha\sigma > 1.2 \\ [\sinh(\alpha\sigma)]^n, & \text{for all } \sigma \end{cases} \tag{7}$$

Therein, $\dot{\varepsilon}$ is the strain rate (s$^{-1}$), $R$ is the universal gas constant (8.31 J·mol$^{-1}$·K$^{-1}$), $T$ is the absolute temperature (K), $Q$ is the activation energy of hot deformation (kJ·mol$^{-1}$), $\sigma$ is the flow stress (MPa), $A$, $n_1$, $n$, $\alpha$, and $\beta$ are the material constants, and $\alpha = \beta/n_1$.

Zener and Hollomon verified that the relationship between stress and strain in isothermal strain depends on the strain temperature and the strain rate [25]. In this study, the hyperbolic sine law was used to investigate the deformation behavior at all stress levels, and it was expressed as a Zener–Hollomon parameter as follows:

$$Z = \dot{\varepsilon}\exp\left(\frac{Q}{RT}\right) = A[\sinh(\alpha\sigma)]^n \tag{8}$$

The following equations can be obtained by taking the logarithms of Equations (7) and (8). $A_1$, $A_2$ are the material constants.

$$\ln\dot{\varepsilon} + Q/RT = \ln A_1 + n_1\ln\sigma \tag{9}$$

$$\ln\dot{\varepsilon} + Q/RT = \ln A_2 + \beta\sigma \tag{10}$$

$$\ln \dot{\varepsilon} + Q/RT = \ln A + n \ln[\sinh(\alpha\sigma)] \tag{11}$$

$$n \ln[\sinh(\alpha\sigma)] - (\ln \dot{\varepsilon} - \ln A) = \frac{Q}{RT} \tag{12}$$

$$\ln Z = \ln A + n \ln[\sinh(\alpha\sigma)] \tag{13}$$

After that, the material constants can be obtained as the average value of the slope obtained by plotting each equation. Equations (14)–(17) are partial differential equations at constant temperature and strain rate. The flow stress used a strain value of 0.15.

$$n_1 = \left. \frac{\partial(\ln \dot{\varepsilon})}{\partial(\ln \sigma)} \right|_T \tag{14}$$

$$\beta = \left. \frac{\partial(\ln \dot{\varepsilon})}{\partial(\sigma)} \right|_T \tag{15}$$

$$n = \left. \frac{\partial(\ln \dot{\varepsilon})}{\partial(\ln[\sinh(\alpha\sigma)])} \right|_T \tag{16}$$

$$Q = \frac{\partial(Rn \ln[\sinh(\alpha\sigma)])}{\partial(1/T)} \tag{17}$$

Figure 5 shows the slopes of $\sigma$, $\ln\sigma$, $\ln\dot{\varepsilon}$, $1000/T$, and $\ln[\sinh(\alpha\sigma)]$ through linear regression analysis, and each data has a linear relationship. $\ln A$ can be obtained from the intercept of the plot of $\ln Z$–$\ln[\sinh(\alpha\sigma)]$.

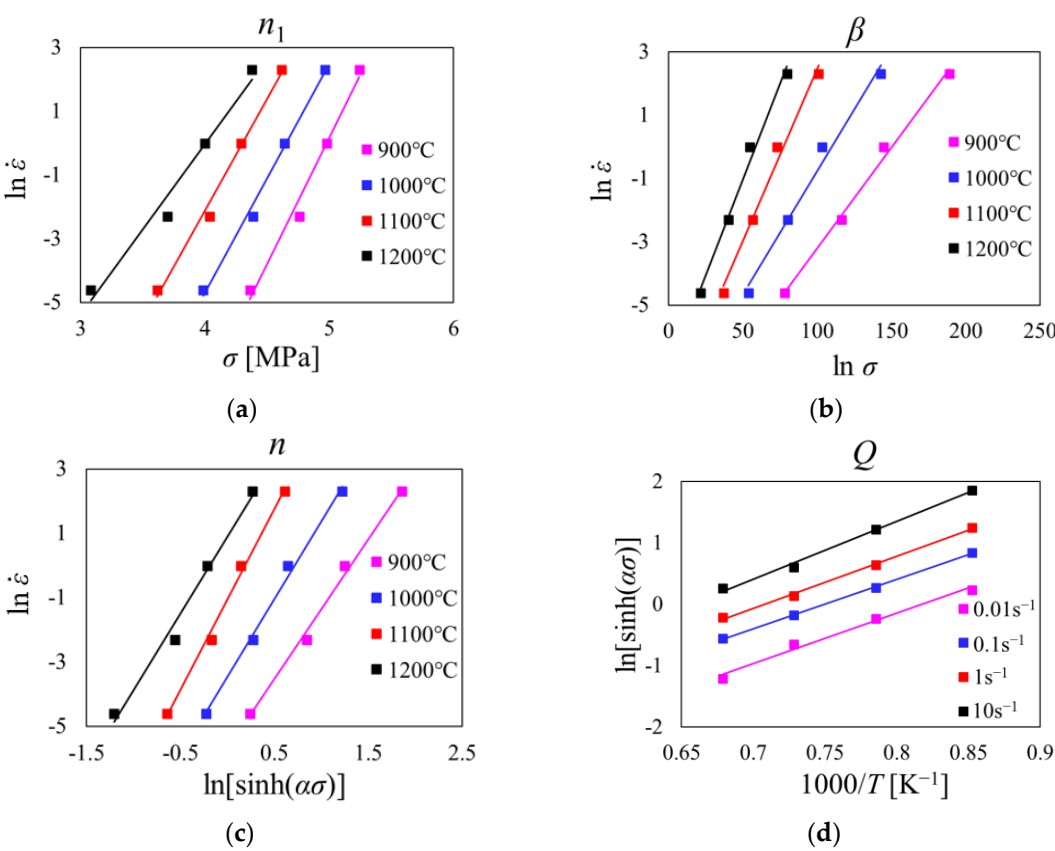

**Figure 5.** *Cont.*

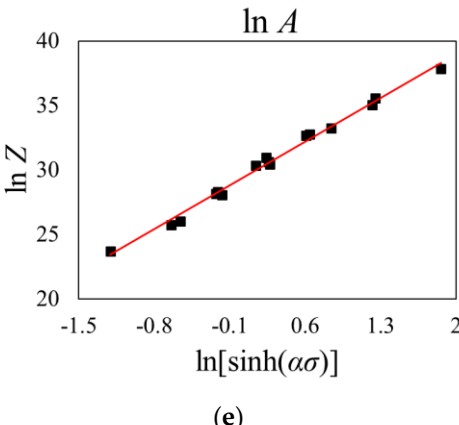

(**e**)

**Figure 5.** Relationships between $\sigma$, $\ln\sigma$, $\ln\dot{\varepsilon}$, $1000/T$, $\ln[\sinh(\alpha\sigma)]$ and $\ln Z$: (**a**) $\sigma$–$\ln\dot{\varepsilon}$, (**b**) $\ln\sigma$–$\ln\dot{\varepsilon}$, (**c**) $\ln[\sinh(\alpha\sigma)]$–$\ln\dot{\varepsilon}$, (**d**) $\ln[\sinh(\alpha\sigma)]$–$1000/T$, (**e**) $\ln Z$–$\ln[\sinh(\alpha\sigma)]$.

Table 2 lists the material constants ($A$, $\alpha$, $n$, $Q$) obtained at strain 0.15. In a previous study, the average activation energy $Q$ of AISI 4340 was reported as 385.584 kJ/mol [2]. Considering that the flow stress is affected by strain, the material constant values were obtained from 0.05 to 1.0 at intervals of 0.05 for each strain.

**Table 2.** Values of material constants at $\varepsilon = 0.15$.

| $\alpha$ (MPa$^{-1}$) | $n$ | $Q$ (kJ/mol) | ln$A$ |
|---|---|---|---|
| 0.013 | 4.89 | 346.79 | 29.29 |

The material constant value according to the strain was obtained using the 7th-order polynomial and expressed as a function of the strain is shown in Figure 6. The coefficients of the fitted polynomial functions are listed in Table 3.

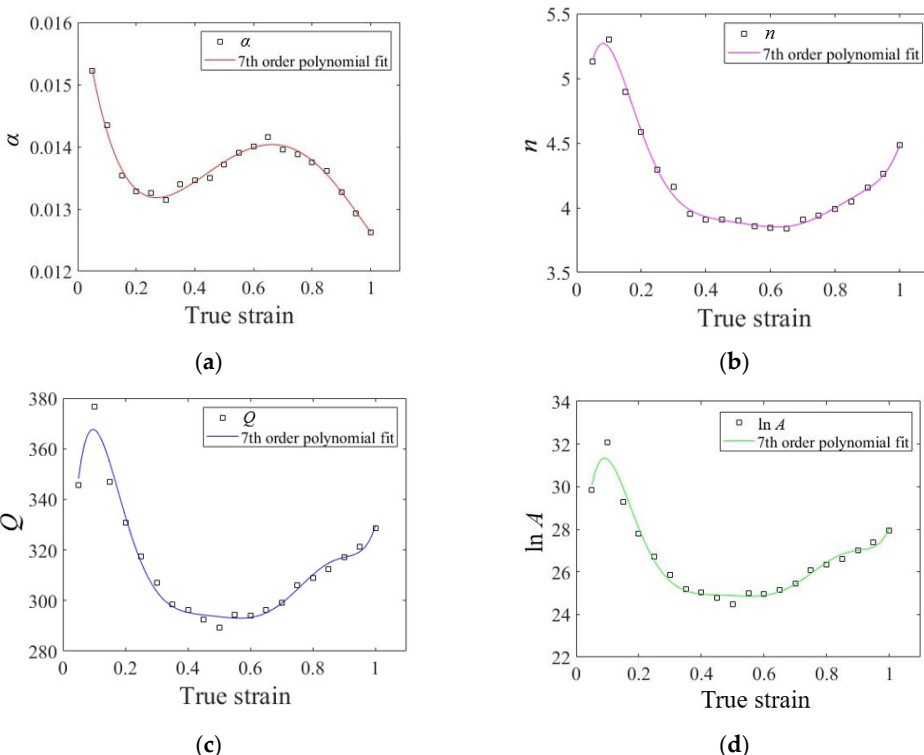

(**a**)

(**b**)

(**c**)

(**d**)

**Figure 6.** The relationship between the material constant value–true strains (**a**) $\alpha$, (**b**) $n$, (**c**) $Q$, and (**d**) ln$A$.

**Table 3.** 7th-order polynomial coefficients of material constants $\alpha$, $n$, $Q$, ln$A$.

| Coefficient | $\alpha$ (MPa$^{-1}$) | $n$ | $Q$ (kJ/mol) | ln$A$ |
|:---:|:---:|:---:|:---:|:---:|
| 1 | 0.0168 | 4.12 | 252 | 22.5 |
| 2 | $-0.0372$ | 35.1 | $3.12 \times 10^3$ | 250 |
| 3 | 0.143 | $-358$ | $-2.93 \times 10^4$ | $-2.44 \times 10^3$ |
| 4 | $-0.274$ | $1.47 \times 10^3$ | $1.20 \times 10^5$ | $1.01 \times 10^4$ |
| 5 | 0.323 | $-3.17 \times 10^3$ | $-2.59 \times 10^5$ | $-2.23 \times 10^4$ |
| 6 | $-0.242$ | $3.75 \times 10^3$ | $3.11 \times 10^5$ | $2.70 \times 10^4$ |
| 7 | 0.0943 | $-2.32 \times 10^3$ | $-1.95 \times 10^5$ | $-1.70 \times 10^4$ |
| 8 | $-0.0118$ | 583 | $4.96 \times 10^4$ | $4.36 \times 10^3$ |

As expressed in Equation (18), the constitutive equation can be expressed as a function of the Zener–Hollomon parameter, and the flow stress was calculated at different strains ranging from 0.05 to 1.0 with an interval of 0.05 with the equation.

$$\sigma = \frac{1}{\alpha} \ln \left\{ \left( \frac{Z}{A} \right)^{\frac{1}{n}} + \left[ \left( \frac{Z}{A} \right)^{\frac{2}{n}} + 1 \right]^{\frac{1}{2}} \right\} \tag{18}$$

Figure 7 shows flow stress calculated through the constitutive equation at temperatures of 900 to 1200 °C and a strain rate of 0.01 to 10 s$^{-1}$. The calculated flow stress showed a large error under the deformation conditions of temperature 900 °C and strain rates of 0.01 s$^{-1}$ and 10 s$^{-1}$, but the error was small under the conditions of 1000 °C or higher. Other alloys reported similar errors. Errors can occur when fitting the material constants. Similar errors were reported equally for other alloys [11,26].

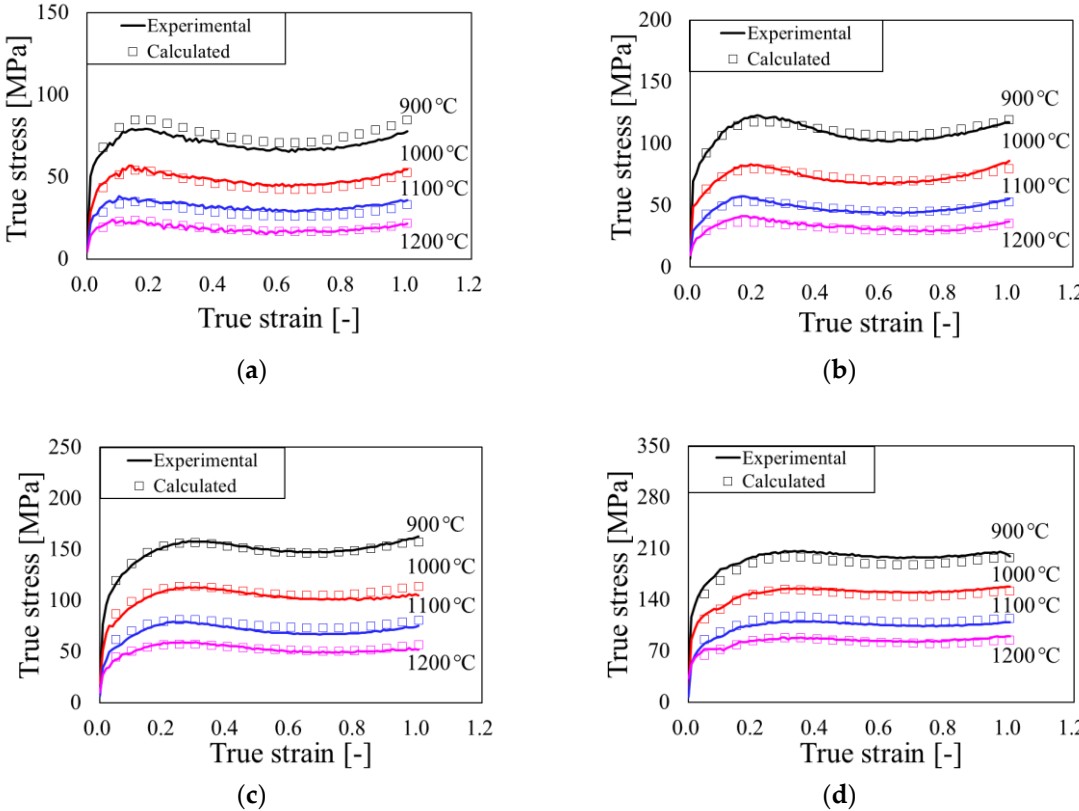

**Figure 7.** Calculated values and experimental values under different deformation conditions are compared. (**a**) 0.01 s$^{-1}$; (**b**) 0.1 s$^{-1}$; (**c**) 1 s$^{-1}$; (**d**) 10 s$^{-1}$.

The accuracy of the calculated flow stress is quantified through the correlation coefficient ($R$) and the mean absolute error ($AARE$) that are calculated by Equations (19) and (20), respectively. $P_i$ is a value calculated through a constitutive equation, and $E_i$ is an experimental value. $\overline{P}$ and $\overline{E}$ are the mean values of the predicted and experimental values, respectively. It can be seen from Figure 8 that there is a good correlation between the calculated and the experimental data. High accuracy was obtained with correlation coefficient ($R$) = 0.9965 and mean absolute relative error ($AARE$) = 4.26%.

$$R = \frac{\sum\limits_{i=1}^{N}(P_i - \overline{P})(E_i - \overline{E})}{\sqrt{\sum\limits_{i=1}^{N}(P_i - \overline{P})^2}\sqrt{\sum\limits_{i=1}^{N}(E_i - \overline{E})^2}} \tag{19}$$

$$AARE(\%) = \frac{1}{N}\sum\limits_{i=1}^{N}\left|\frac{E_i - P_i}{E_i}\right| \times 100\% \tag{20}$$

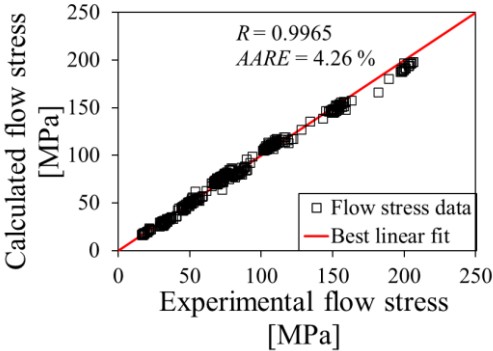

**Figure 8.** Correlation between experimental and predicted flow stress by the constitutive equations.

*3.3. Processing Map*

3.3.1. Processing Map Theory

In order to construct a processing map based on the dynamic material model (DMM), the total input energy $P$ that dissipated at any instant comprises the $G$ content that is dissipated by plastic deformation, and $J$ co-content, which is the power dissipated by changes in microstructure [27–30]. The flow stress between the hot deformation of the metal can be expressed as Equation (21), which is a function of the strain rate at a constant temperature.

$$\sigma = K\dot{\varepsilon}^m \tag{21}$$

where $m$ is a strain rate sensitivity coefficient, $K$ is the strength constant that depends on the temperature. The total energy ($P$) received by a material is divided into two energies [31–34]. The total energy $P$ may be expressed as the sum of the terms $G$ and $J$ as:

$$P = \sigma\dot{\varepsilon} = \int_0^{\dot{\varepsilon}}\sigma d\dot{\varepsilon} + \int_0^{\sigma}\dot{\varepsilon}d\sigma = G + J \tag{22}$$

The strain rate sensitivity in Equation (21) can be expressed as Equation (23), and $G$ and $J$ can be expressed with the strain rate sensitivity [20,35]. $m$ is according to each strain and temperature, and the condition for a stable flow is $m > 0$. Total energy ($P$) can be expressed by combining Equations (21) and (23).

$$\frac{dJ}{dG} = \frac{\dot{\varepsilon}d\sigma}{\sigma d\dot{\varepsilon}} = \frac{\dot{\varepsilon}\sigma d\ln\sigma}{\sigma\dot{\varepsilon}d\ln\dot{\varepsilon}} \approx \frac{\Delta\log\sigma}{\Delta\log\dot{\varepsilon}} \equiv m \tag{23}$$

$$G = \frac{\sigma \dot{\varepsilon}}{1 + m}, \ J = \frac{m \sigma \dot{\varepsilon}}{1 + m} \tag{24}$$

The efficiency of power dissipation ($\eta$) represents the relative efficiency of a material dissipating energy due to internal microstructure change, and it is defined as Equation (25) [21]. It shows the efficiency of microstructure change through contour lines in the processing map. For ideal plastic flow ($m = 1$), the value of J reaches its maximum value.

$$\eta = \frac{J}{J_{\max}} = \frac{\int_0^{\dot{\varepsilon}} \sigma d\dot{\varepsilon}}{\sigma \dot{\varepsilon}/2} = \frac{2m}{m + 1} \tag{25}$$

Based on DMM and the extremum principles of irreversible thermodynamics as applied to large plastic flow [36], Kumar [37] and Prasad [38] developed an instability criterion for predicting flow instability ($x$) as follows.

$$\xi(\dot{\varepsilon})_{\text{Kumar-Prasad}} = \frac{\partial \ln(m/m + 1)}{\partial \ln \dot{\varepsilon}} + m < 0 \tag{26}$$

The plastic instability factor is the criterion for determining the plastic instability condition according to the temperature and strain rate given through the flow instability map. Taking use of the principle of the maximum rate of entropy of production, a continuum criterion for the occurrence of flow instabilities is defined in terms of another dimensionless parameter ($\xi$). In the region corresponding with a negative plastic instability factor value ($\xi < 0$), instable plastic behavior may occur during plastic deformation [39].

### 3.3.2. Analysis of Processing Map

The flow stress data were obtained to evaluate different strain rates and temperatures at a particular strain. Further, to compute the strain rate sensitivity ($m$) values, the experimental data were plotted as log($\sigma$) vs. log($\dot{\varepsilon}$) at each temperature, as shown in Figure 9a. These plots at each temperature were fitted with cubic splines, as shown in Figure 9b, at the true strain of 0.5. According to Equation (22), the first derivation of cubic spline was taken to calculate m values at small intervals of strain rates and temperatures.

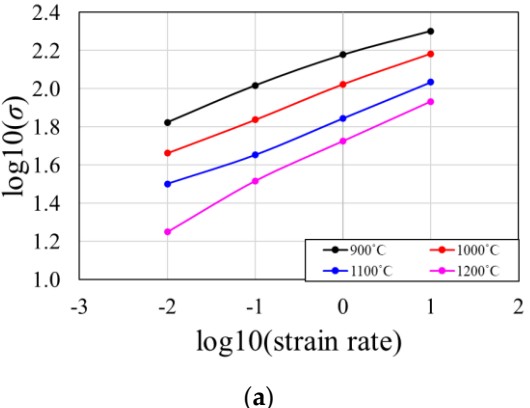
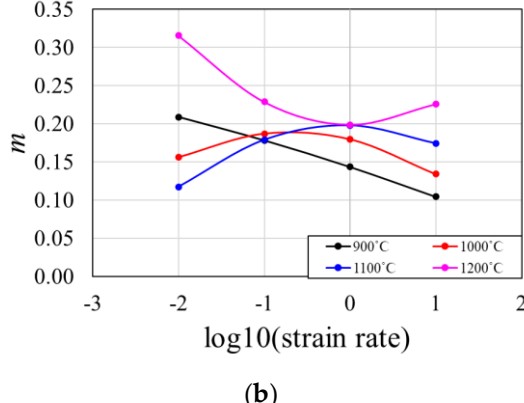

(a)  (b)

**Figure 9.** Interpolated curves to describe the relationship at strain 0.5. (**a**) between log($\sigma$) and log($\dot{\varepsilon}$), (**b**) between strain sensitivity ($m$) and log($\dot{\varepsilon}$).

The processing map can be constructed by superpositioning the flow instability map on the power dissipation efficiency map. Figures 10 and 11 show the treatment map at 0.1 intervals from 0.1 to 1.0 strain through the flow stress before and after correction for friction and temperature change. It can be confirmed that the tendency of the processing map through two flow stresses is similar before strain 0.4, but the tendency changes after strain 0.4. The effect of friction and temperature of flow stress is evident from strain 0.4 or higher. This stress change results in a strain rate sensitivity ($m$) change. In the processing

map, colored regions indicate flow instability that needed to be avoided. A high value of the contour region is consuming more energy for microstructural evolution. The region excluding color is a stable region with good workability.

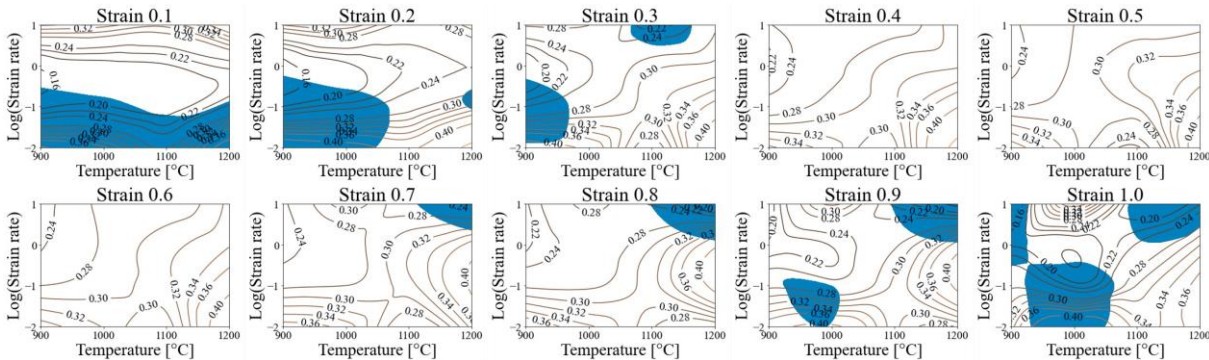

**Figure 10.** Experimental flow stress processing map.

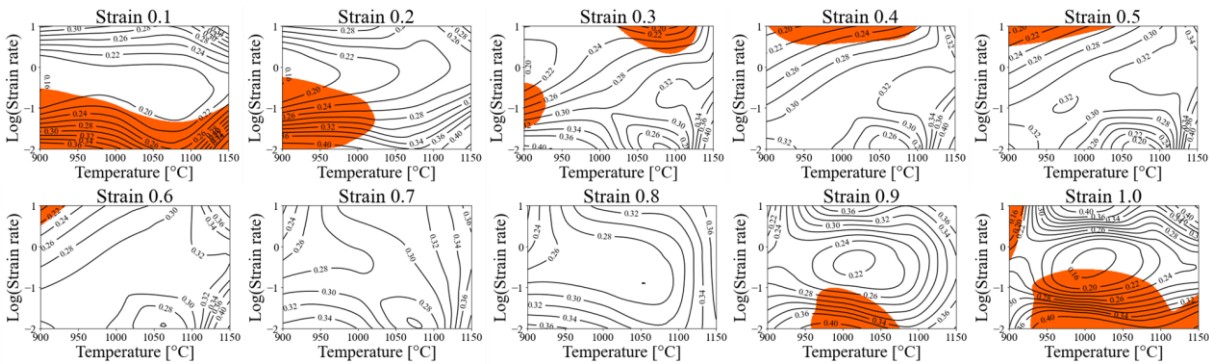

**Figure 11.** Corrected flow stress processing map.

Figure 12 shows the power dissipation efficiency value of 0.4 or more at low strain rates of 0.01 s$^{-1}$ and high temperatures of 1150 °C. At strains 0.1–0.3 and 0.3–0.6, the instability zone shifts to a lower temperature with increasing strain. At strains 0.1–0.2 and 0.9–1.0, the region of power dissipation efficiency value of 0.4 or over overlaps the region of instability. The flow instability zone was not found at strains of 0.7–0.8, indicating that it is stable. As shown in Figure 12a,b, processability may be determined by stacking the 3D processing map at 0.01 intervals from strain 0.1 to 1.0 [40]. According to the color depth, Figure 12a represents power dissipation efficiency, and Figure 12b flow instability.

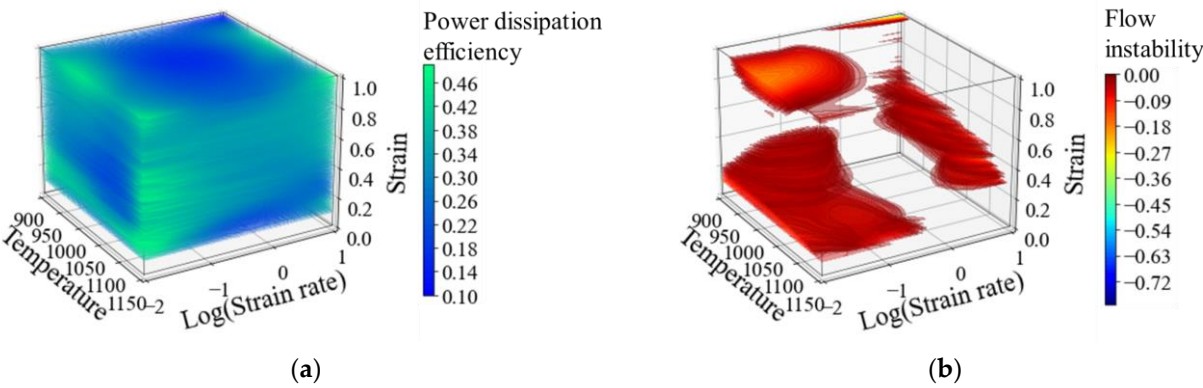

**Figure 12.** 3D processing map. (**a**) 3D power dissipation efficiency map, (**b**) 3D flow instability map.

The existing 2D processing map has limitations in determining the flow instability and power dissipation efficiency according to the temperature, strain rate, and strain change

during plastic deformation. In addition, there is a limit to understanding the deformation behavior of the product in the actual production process. The advantage of the developed 3D processing map is the immediate identification of instability zones and the power dissipation efficiency distribution. FEM analysis can be used to display strain, strain rate, and temperature data for each node of a deformable material in a 3D processing map. Therefore, it is possible to avoid the flow instability zones and control the temperature and speed of the forging process overlapping with high power dissipation efficiency zones.

### 3.4. Hot Forging Process Temperature and Speed Controls Based on a 3D Processing Map

Figure 13 illustrates the FEM simulation under similar conditions to the compression test conducted to obtain the proposed 3D processing map. Since the forging process has negligible elastic deformation compared to the amount of plastic deformation, a rigid-plastic finite element analysis was performed through the commercial finite element analysis software Simufact Forming 2022 (2022, Hexagon, Stockholm, Sewden). The process speed is shown in two cases in Figure 13c. Figure 14 compares the data pattern according to temperature and process speed.

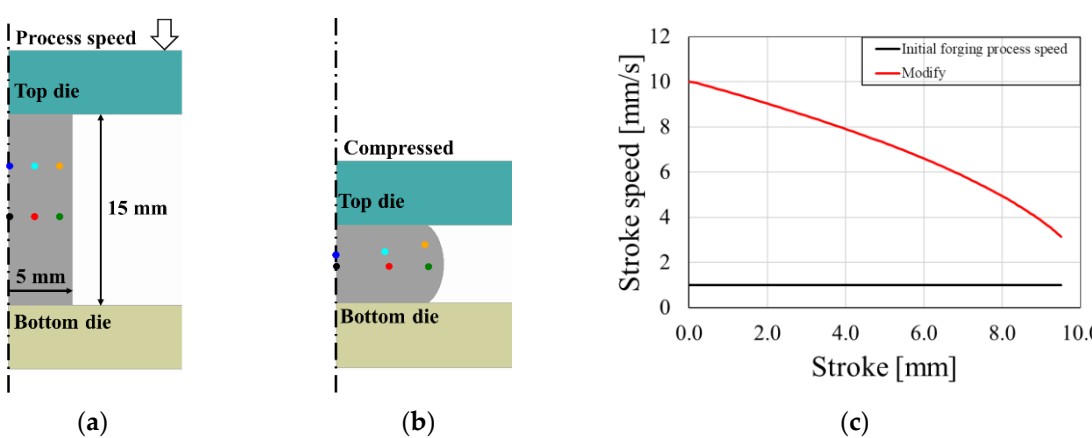

**Figure 13.** Schematic of axisymmetric cylinder forging simulation: (**a**) initial, (**b**) compressed, (**c**) process speed.

In the FEM simulation, the top and bottom die is the rigid body, and a constant coulomb friction coefficient model was used to represent the friction between the dies and the workpiece, with a friction factor of 0.3. The number of mesh in the workpiece is 20,750. The heat transfer coefficient to the environment (HTC) and the heat transfer coefficient to the workpiece was set to 20 W/(m$^2$·K). The 3D processing map is a space composed of strain, strain rate, and temperature. A node at each location in the 3D flow instability map space can be plotted according to deformation history using particle tracking techniques. The operator can thus control the temperature and process speed of the material and die in hot forging by viewing the deformation history plotted on the map.

As shown in Figure 13, the process speed was 1 mm/s, the initial temperature of the specimen was 1000 °C, the top and bottom die temperatures were 950 °C, and the specimen was compressed to a true strain of 1.0. As shown in Figure 13a, the strain, strain rate, and temperature data through particle tracking techniques in the specimen pass through the flow instability zones.

Figure 14a shows that in order to be able to control the particle data to avoid flow instability zones through the 3D processing map, the process needs to be carried out at a higher temperature and a higher speed. Therefore, as shown in Figure 13c, the process speed is modified as gradually decreased from 10 mm/s to 3.13 mm/s. The initial specimen top and bottom die temperatures were increased by 50 °C. Figure 14b,c show that the particle data up to true strain 1.0 avoid the instability zone and overlap the power dissipation efficiency zone of 0.32 or more. The high power dissipation efficiency indicates that the material dissipates more energy for the microstructural changes. This

determines the best process parameters recommended for designing and conducting hot forging processes [41–43].

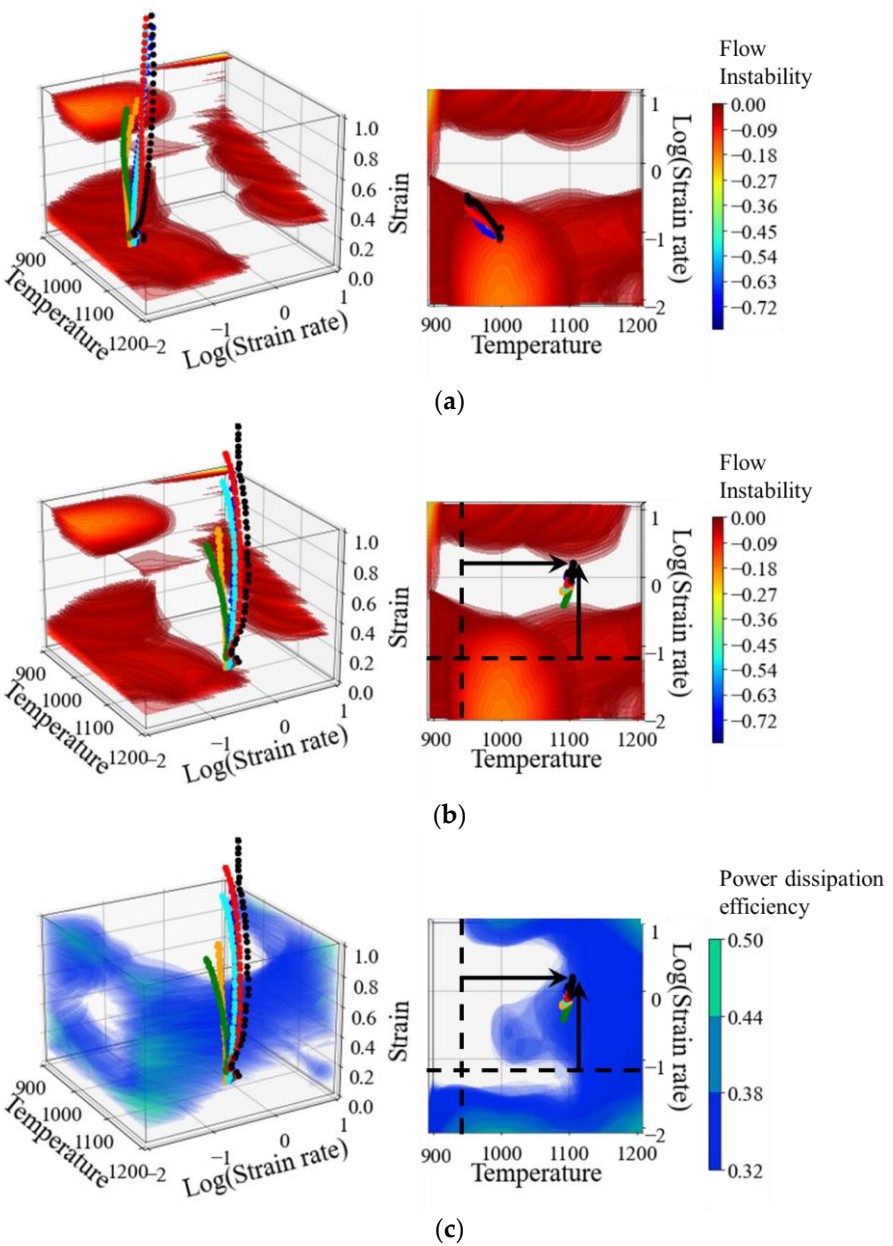

**Figure 14.** Control of process temperature and speed using a 3D processing map in hot forging. (**a**) The initial process, (**b**) modified process on the 3D flow instability map, (**c**) modified process on the 3D power dissipation efficiency map.

*3.5. Microstructure Analysis*

Microstructure Analysis of Processing Map

The optimal hot workability determined by the Processing map is related to the power dissipation efficiency and should avoid the flow instability. The forming defects of microstructures that may occur in the unstable area include adiabatic shear bands, flow localization, dynamic strain ageing (DSA), and kink bands [44].

Figure 15 shows the microstructure by photographing the cross-section of the compressed specimen, as shown in Figure 1c, by each deformation condition with EBSD. Thermal energy relaxes the internal strain energy during high-temperature strain. Because this promotes the migration of grain boundaries, the grain size increases with increasing

temperature. On the other hand, when the strain is increased, the dislocation density and internal strain energy increase, which promotes DRX and shortens the grain growth time to reduce the grain size. The red border indicates the microstructure corresponding to the instability zones of the processing map. As a result of observing the microstructure corresponding to the instability zone, at 900 °C DRX occurs less, and a shear band is observed. Partial recrystallization occurred in some areas, as indicated by the dashed yellow line at 1000 °C. It indicates that an inhomogeneous microstructure is obtained under the corresponding deformation conditions. Flow localization is found in the 1100–1200 °C region. Meanwhile, the stable zones with a temperature of 1100–1200 °C and a strain rate of 0.01–1 s$^{-1}$ have an elongated needle-like ferrite structure. Acicular ferrite structure increases toughness [45–47]. Therefore, it can be seen that toughness is advantageous due to the needle-shaped ferrite structure when deformed under the corresponding conditions.

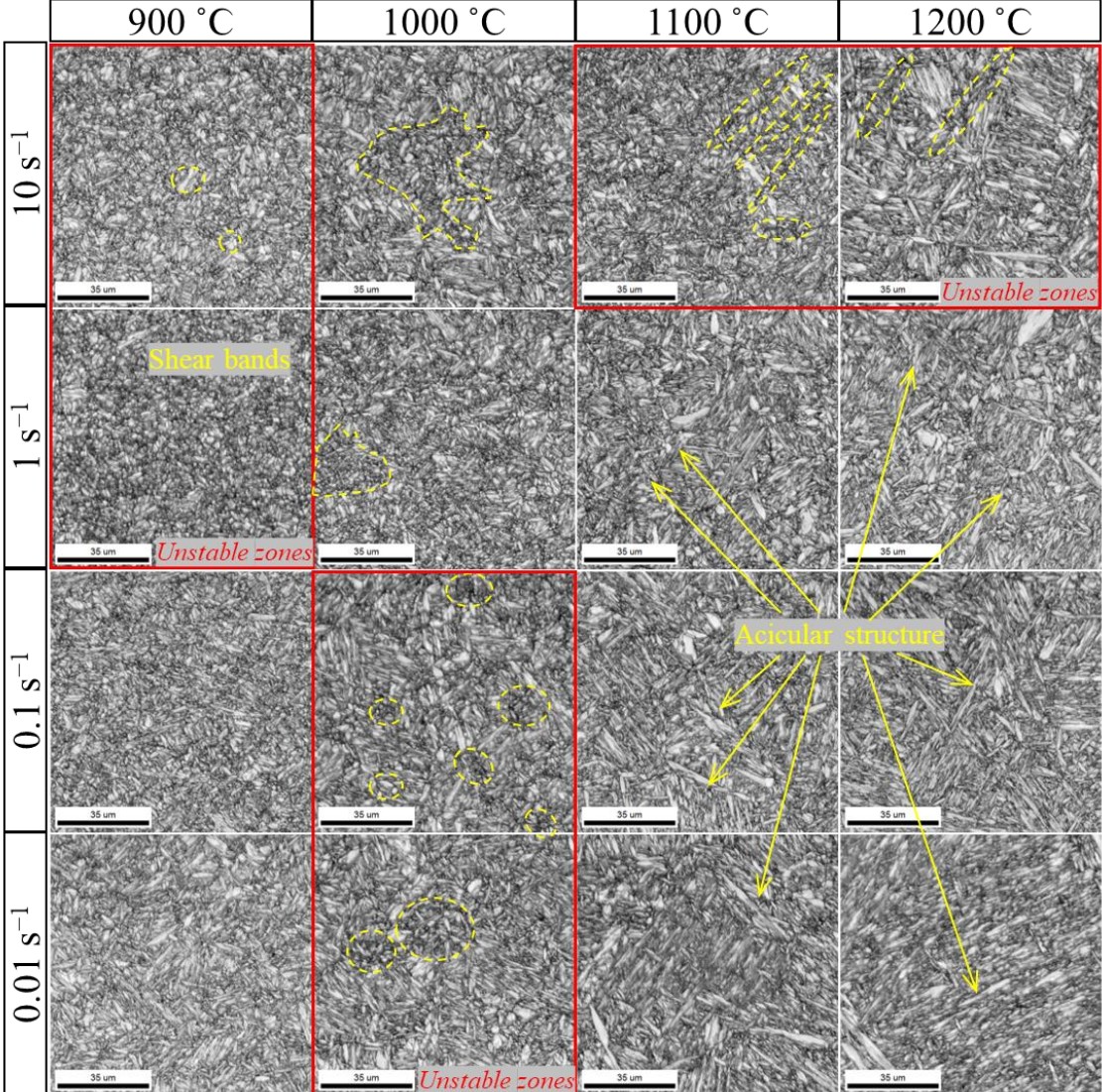

**Figure 15.** The image quality map of the specimen compressed at different conditions.

## 4. Conclusions

The conclusions obtained using the constitutive equation and 3D processing map are as follows.

The constitutive equation of AISI 4340 steel was determined through the relationship between the Arrhenius model and the Zener–Hollomon (*Z*) parameter and showed a high prediction rate with correlation coefficient (*R*) = 0.9965 and mean absolute error (*AARE*) =

4.26%. The small error indicates that the above model can well describe the hot deformation behavior of AISI 4340 steel studied in this paper.

Through the 3D flow instability map stacked at 0.01 intervals from strain 0.1 to 1.0, it is possible to check the process conditions that can avoid unstable regions visually. The process conditions are temperature 900–1200 °C, strain rate 0.55–1.9 s$^{-1}$ region, and temperature 900 °C, strain rate over 0.55 s$^{-1}$ stable in the region. Therefore, the 3D flow instability map should induce process conditions (process temperature and speed) into the unpainted area.

From strain 0.4 or more, the highest power dissipation efficiency value is 0.55 at temperature 1200 and strain rate 0.01 s$^{-1}$. The high power dissipation efficiency can promote the occurrence of dynamic recrystallization.

In the hot forging deformation simulation, particle tracking technology can plot strain, strain rate and temperature data on the 3D processing map to control process temperature and rate, avoid unstable regions, and achieve process conditions, including regions with high power dissipation efficiency.

**Author Contributions:** Conceptualization, J.P. and Y.K.; methodology, J.P. and Y.K.; software, J.P.; validation, J.P., Y.K. and N.K.; formal analysis, J.P. and Y.K.; investigation, Y.K.; resources, N.K. and S.S.; data curation, J.P. and Y.K.; writing—original draft preparation, J.P. and Y.K.; writing—review and editing, J.P., Y.K. and N.K.; visualization, J.P. and Y.K.; supervision, N.K.; project administration, N.K.; funding acquisition, N.K. All authors have read and agreed to the published version of the manuscript.

**Funding:** This research was funded by Ministry of Trade, Industry & Energy (MOTIE) of Korea, grant number 20017503 and 20013060.

**Institutional Review Board Statement:** Not applicable.

**Informed Consent Statement:** Not applicable.

**Data Availability Statement:** Not applicable.

**Acknowledgments:** This work was supported by the Material Component Technology Development Program (20017503) funded by the Ministry of Trade, Industry & Energy (MOTIE, Korea) and the Material Component Technology Development Program (20013060) funded by the Ministry of Trade, Industry & Energy (MOTIE, Korea).

**Conflicts of Interest:** The authors declare no conflict of interest.

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
