# Peer review of "Characterization of Hot Workability in AISI 4340 Based on a 3D Processing Map"

_metals, doi:10.3390/met12111946_

Round 1

Reviewer 1 Report

In this paper, the author investigated the DRXed behavior and obtained the process map of AISI4340 alloy. The results are of scientific meaning while there are some specific issues to be solved before it can be accepted the academic society.  

(1) The English writing of this manuscript should be improved. For instance, at line 56, various alloy steel should be “steels”. Please have a careful check about this grammar mistakes.

(2) Please add the full name of EBSD in abstract.

(3) “In this study, a method of controlling the temperature and speed of the hot forging 65 process to prevent defects is present by visualizing the deformation history of the nodes 66 inside the material on the 3D flow instability map space. The compression test results”. This sentence is not readable. Please rewrite it,

(4) Please add “a, b, and c” to the pictures of figure 1, to better illustrate the left is before compression, the middle is after compression, and the right one is how the sample is taken for observation.

(5)  Please add the composition of AISI4340 steel that used in this study.

(6) Please add the equipment used for EBSD characterization and how the data was processed.

(7) Please add the min and max of each color in GOS map.

(8) The author indicated partial DRX at 1000c.While, at line 346, the author indicate complete DRX happened. Please double check the EBSD results. It is clear from Figure 13 that they are partial DRXed structure.

(9) For the DRXed grain size, it is clear that the increased temperature leads to increased DRXed grains. The DRXed grain size is closely related to the immigration of grain boundaries during the DRXed process. And the ability of grain boundary is related to the pinning effect and its mobility. Please check about the related refs (https://doi.org/10.1016/j.msea.2021.142143) and add explain about it. 

Reviewer 2 Report

Comments:

This manuscript presented a method to characterize stability zones as a function of strain rate and temperature for hot compression tests of the AISI4340 steel. Although this work seems to be interesting, there are many ambiguities in using terminologies and presenting equations, causing many confusions in understanding the text and making some of the discussions/conclusion not convincing. This manuscript has to be reworked and major revisions need to be done. Specific comments are as follows:

1. The use of ‘processing map’ is confusing , change to something like “”.

2. Line 56, various alloyed steels.

3. Compositions of the AISI 4340 steel ?

4. Section 3.1 should be the method. All the descriptions for the equation are very unclear, please describe them better and present the meaning of each symbol.

5. Equation 5, is RM measured after the hot compression test ? Otherwise, it is not clear how you can calculate RT.

6. Section 3.2 equations and symbols are also very unclear. For example, Z is Zener-Hollomon parameter but is not explained.

7. Equation 7 is wrong, because it only corresponds to the third scenario of equation 6. It should be rewritten as .

8. Line 146-147 is redundant because the meanings of R, T and Q have been explained in line 137-139.

9. Equations 9-13 should be re-described! What are A1, A2. The link to equation 6 and 7 should be more clear.

10. Mixed use of ‘power’ and ‘energy’ should be avoided. Unified to use “energy”.

11. Line 207, clarify that the values of m is between 0 and 1.

12. Equation 24, the first part is the same as equation 22.

13. Not clear how the authors can correlate the temperature and strain rate with η. The hot compression was only performed at temperatures of 900, 1000, 1100 and 1200 °C. Even there are about 10-15 °C variance for different strain rates, how you can construct a continuous ‘processing map’ to show a full range of temperature influence? Is this done by interpolation for such a big temperature gap (~80 °C)? This result is not convincing.

14. What is the physical mechanism of equation 26 to extract the instability region? It looks rather empirical, above all the empirical equation presented in the whole manuscript.

15. Line 230-237, comparing Figures 8 and 9, the maps are very different already starting from a strain of 0.4, rather than 0.6.

16. ‘Power dissipation efficiency’ is very confusing. Can you change to microstructural dissipation fraction?

17. ‘Processing map’ is also very confusing. This is a diagram showing the correlation among temperature, strain rate and microstructural dissipation fraction. Please consider to use a better word / name.

18. Line 304, ‘particle data’ suddenly pops up and it was never explained before.

19. Figure 13, it is really unconvincing because the morphologies of these ferrites do not show a significant difference between the unstable and stable zones.

20. Figure 14, what are the meanings of the colors. I was completely lost when I was reading line 339-350.

21. How you identify DRX?

22. Line 378. According to experimental data, isn’t it the smallest of D_DRX appear at a temperature of 900C and a strain rate of 0.1 s-1? And the largest appears at a temperature of 1200C and a strain rate of 0.01 s-1?

23. Table 3. GOS map? What is this? Area fraction of the grains with orientation spread smaller than 5°?

24. The Conclusion section needs to be rewritten.

25. English has to be polished extensively.

Reviewer 3 Report

The obtaining of the constitutive equations and determining the optimal deformation conditions are necessary for the development of deformation technologies for the steel. The authors of the paper "Characterization of hot workability in AISI 4340 based on a 3D processing map" have made very interesting work for the determination of the hot deformation ability of the steel. The authors determine the stable and unstable deformation range using the processing maps and optimize the forging process by applying the FE simulation. Despite of the good presentation of the obtained results, the paper should be significantly improved before the next submission. The authors should consider the following comments to improve the manuscript:

1. The main problem of the paper is that the authors did not correctly consider the influence of friction and adiabatic heating during the deformation. Despite of the provided method of the calculations the stress-strain curves are not correspondent to the real hot deformation process. It is hardly that the steady-state stage is absent on the stress-strain curves. There are no microstructural reasons for increases in the stress after about 0.4. The increase is due to friction. As a result, all coefficients in the constitutive model and constructed processing maps are incorrect. The authors should recalculate the stress-strain curves considering friction and adiabatic heating and reconstruct the constitutive model and processing maps. It is recommended to apply the method for correction from [10.1016/j.jallcom.2018.08.010, 10.1134/S0031918X14080031].

2. Processing maps for the investigated steel were constructed previously in 10.4028/www.scientific.net/AMR.683.301 and 10.1016/j.jmrt.2019.05.018. What is the difference in the presented and previous results? It is recommended to add a comparison of the processing maps and effective activation energy values with obtained previously.

3. The authors should provide the chemical composition of the investigated alloy. Also, it is recommended to provide the initial microstructure of the steel before deformation including the austenite grain size.

4. The modeling of the rain size evolution seems to be incorrect. The deformation was proceeded in the austenite phase range. However, the authors using EBSD techniques have analyzed the martensite microstructure obtained by the air-quenching after hot deformation. It is strongly recommended to remove the 3.5.2 Part from the manuscript to prevent the misleading of the readers.

5. The additional information about FE modelling (number of the elements, boundary conditions, etc.) should be added to the manuscript.

6. Minor correction:

- Line 231: the intervals should be 0.1 but not 0.01.

- Large angle grain boundaries should have misorientation angle larger the 15° but not lower than 5° as authors have written in the line 343.

Round 2

Reviewer 1 Report

The mentioned issues have been solved. It is suggested to be accepted after minor English checking.  

Author Response

Minor English grammar has been revised. Thank you.

Reviewer 2 Report

Further corrections are required:

1) Descriptions of Eqs. 8-12 are still unclear. Eq. 8 and 9 are logarithms of Eq. 6, 7 respectively, while Eq. 10-12 corresponds to Eq. 7. Please revise carefully !

2) Add the response to my question on constructing a continuous processing map to the text.

3) Also add the response to my question on the physical meaning of the equation for deriving the instability region into the text.

4) Although the 'processing map' is extensively used in hot deformation, you should point out this is a special terminology and cite the original paper which proposed this word.

Reviewer 3 Report

The authors of the paper "Characterization of hot workability in AISI 4340 based on a 3D processing map" have partially answered the previous comments. The manuscript is still needed to be improved accordingly following comments:

1. A comparison of the processing maps and effective activation energy values with obtained previously for the investigated steel was not provided.

2. The friction correction does not negotiate the influence of the friction. There is not any microstructural evidence for increases in the stress at high strain values.

3. The number of the digits after a dot for the constant values and microstructural characteristics should be decreased accordingly the value of the error. The standard deviations should be added to the values of the microstructural characteristics.

4. The caption for Figure 11 is not correct. 3D processing maps assume changing in strain.

Round 3

Reviewer 3 Report

The authors have answered the previous comments and improved the manuscript. Te paper may be accepted for publication.